# A Systematically Assembled Signature of Genes to be Deep-Sequenced for Their Associations with the Blood Pressure Response to Exercise

**DOI:** 10.3390/genes10040295

**Published:** 2019-04-11

**Authors:** Linda S. Pescatello, Paul Parducci, Jill Livingston, Beth A. Taylor

**Affiliations:** 1Department of Kinesiology, University of Connecticut, Storrs, CT 06269, USA; Paul.Parducci@uconn.edu (P.P.); Beth.Taylor@uconn.edu (B.A.T.); 2Institute for Systems Genomics, University of Connecticut, Storrs, CT 06269, USA; 3Homer Babbidge Library, Health Sciences, University of Connecticut, Storrs, CT 06269, USA; Jill.Livingston@uconn.edu; 4Preventive Cardiology, Hartford Hospital, Hartford, CT 06269, USA

**Keywords:** antihypertensive therapy, healthy lifestyle, physical exercise

## Abstract

*Background*: Exercise is one of the best nonpharmacologic therapies to treat hypertension. The blood pressure (BP) response to exercise is heritable. Yet, the genetic basis for the antihypertensive effects of exercise remains elusive. *Methods:* To assemble a prioritized gene signature, we performed a systematic review with a series of Boolean searches in PubMed (including Medline) from earliest coverage. The inclusion criteria were human genes in major BP regulatory pathways reported to be associated with: (1) the BP response to exercise; (2) hypertension in genome-wide association studies (GWAS); (3) the BP response to pharmacotherapy; (4a) physical activity and/or obesity in GWAS; and (4b) BP, physical activity, and/or obesity in non-GWAS. Included GWAS reports disclosed the statistically significant thresholds used for multiple testing. *Results:* The search yielded 1422 reports. Of these, 57 trials qualified from which we extracted 11 genes under criteria 1, 18 genes under criteria 2, 28 genes under criteria 3, 27 genes under criteria 4a, and 29 genes under criteria 4b. We also included 41 genes identified from our previous work. *Conclusions:* Deep-sequencing the exons of this systematically assembled signature of genes represents a cost and time efficient approach to investigate the genomic basis for the antihypertensive effects of exercise.

## 1. Introduction

Cardiovascular disease (CVD) is the leading cause of death worldwide, claiming 31% (17.5 million) of deaths globally [1,2]. Hypertension is the most common, costly, and preventable CVD risk factor [1]. The American College of Cardiology (ACC)/American Heart Association (AHA) recently redefined hypertension to a lower blood pressure (BP) threshold of 130 mmHg for systolic BP (SBP) or 80 mmHg for diastolic BP (DBP) [3]. This change has been met with some opposition. Nonetheless, the lower BP thresholds now classify 50% of adults in the United States with hypertension [1], underscoring the importance of hypertension as a public health problem. 

The authors of the AHA/ACC report rated exercise as one of the best nonpharmacologic therapies to treat hypertension because aerobic exercise training lowers BP 5–8 mmHg among adults with hypertension [3]. The magnitude of these BP reductions rival those that result from taking antihypertensive medication [4,5], may lower the risk of CVD risk by 4–22% and stroke by 6–41% [6,7,8], and reduce the resting BP of some adults with hypertension into normal ranges [9,10]. Accordingly, professional organizations from around the world recommend adults with hypertension participate in 30–60 min/day of aerobic exercise, such as walking or jogging, on most days of the week [5,11].

Despite the well-documented antihypertensive benefits of exercise, many adults with hypertension do not exercise to lower their BP [12,13]. Reasons for this non-adherence are varied; however, one reason relevant to the topic of this systematic review is that there is significant inter-individual variability in the BP response to exercise partially attributed to genetic predispositions that led some to believe exercise does not work as antihypertensive therapy [14,15]. Indeed, investigators from the *HE*alth, *RI*sk Factors Exercise *T*r*A*ining and *GE*netics or *HERITAGE* Family Study involving over 700 subjects established the BP response to aerobic exercise is heritable (*h* = 0.13–0.42) [14,16,17,18,19]. Nearly 20 years ago this discovery prompted our laboratory group [20,21,22,23,24] and others [16,17,18,19,25] to conduct candidate genes association studies to identify genetic variants that account for a clinically meaningful proportion of the variability in the BP response to exercise. However, these efforts have met with little success [17,25,26].

More recently, we exploited advances in genomic technology that emerged since our discovery phase candidate gene association studies as well as other strategies to bolster our statistical power to detect genetic variants associated with the BP response to exercise in a replication cohort of subjects with hypertension whose characteristics resembled those from our earlier studies. In this series of studies, we deep-sequenced the exons of genes using the Illumina TruSeq Custom Amplicon kit (Catalog# FC-130-1001, Illumina, San Diego, CA, USA) on a prioritized signature of 41 BP and exercise genes that contained genes identified from our prior work in addition to those obtained from a systematic review of the literature of genes reported to be associated with hypertension or the BP response to exercise or pharmacotherapy [27,28,29]. After adjustment for multiple testing, despite the small sample size, we found variants in 61% of the 41 genes in the prioritized panel associated with the BP response to exercise. We attributed the high proportion of the significant BP-genotype associations that we found to a focused inquiry of variants with a gene signature obtained from a systematic review of the literature that reduced the search space within the genome; and the use of high throughput exon sequencing to concentrate on functional gene regions and standardized BP and exercise protocols that were well controlled and closely supervised.

The purpose of this systematic review is to update and expand our original systematic review to assemble a prioritized signature of BP and exercise genes whose exons can then be deep-sequenced among a larger, more ethnically and gender diverse sample of adults that were reported in the qualifying studies to have hypertension to better inform the genomic basis for the antihypertensive effects of exercise. The long-term goal of this work is to develop personalized exercise prescriptions based upon genetic predispositions and other clinical characteristics to optimize the BP benefits of exercise.

## 2. Systematic Review Methods

This systematic review followed the specifications of the Preferred Reporting Items for Systematic Reviews and Meta-Analyses (PRISMA) statement [30,31]. A series of comprehensive Boolean searches were run in PubMed (including Medline). The first search replicated the original search done by Bruneau et al. [25] in the four databases of Medline, Biosis, Scopus, and Web of Science and included contiguous dates. The additional three searches were run from earliest coverage to 11 May 2017 to locate trials that met our predetermined inclusion criteria. These *a priori* criteria were human genes in major BP regulatory pathways reported to be associated with: (1) the BP response to exercise; (2) hypertension in genome-wide association studies (GWAS); (3) the BP response to pharmacotherapy; (4a) physical activity and/or obesity, a major risk factor for hypertension as 80% of adults with hypertension are overweight to obese [32] in GWAS; and (4b) BP, physical activity, and/or obesity in non-GWAS from relevant reference searches of the authors’ files. We also included the genes from the prioritized signature of BP and exercise genes from our previous work [27,28,29].

The major regulatory BP pathways that a qualifying gene could be in were the renin angiotensin aldosterone system, endothelial nitric oxide synthase pathway, and/or the sympathetic nervous system, and/or pathways involved with vascular function and structure, fluid and electrolyte control, inflammation, insulin sensitivity, lipid metabolism, and obesity. Of note, many of the GWAS we initially located as indicating they involved genes associated with hypertension upon closer scrutiny involved genes associated with resting BP rather than hypertension *per se* so that they were excluded. In all GWAS for the gene to be included on the panel, the reported associations with the BP response to exercise must have met preestablished statistically significant thresholds for multiple testing which is commonly set at *p* = 5 × 10^−8^. The full search strategy for each of the four *a priori* criteria are depicted in Table 1, Table 2, Table 3 and Table 4.

Potential reports were screened by PP for title and abstract, and by PP and LSP for full text review to determine if they qualified. In addition, the authors performed manual searches of reference lists from relevant reports in their files for possible inclusion. Any questions regarding qualification were resolved by discussion by PP and LSP.

## 3. Results

The search yielded 1408 reports plus 14 records obtained from manual searches of the authors’ files. Of these, 57 trials qualified. Please see Figure 1 for the flowchart detailing the systematic search of potential reports and selection process for the reports that qualified. Of the qualifying trials containing human genes in major BP regulatory pathways, we extracted 11 genes associated with the BP response to exercise under criteria 1 [33,34,35,36,37,38,39,40]; 18 genes associated with hypertension in GWAS under criteria 2 [41,42,43,44,45,46,47,48]; 28 genes associated with the BP response to pharmacotherapy under criteria 3 [49,50,51,52,53,54,55,56,57,58,59,60,61,62,63,64,65,66,67,68,69,70]; 27 genes associated with physical activity and/or obesity in GWAS under criteria 4a [71,72,73,74,75,76,77,78,79,80,81,82,83,84,85]; and 29 genes associated with BP, physical activity, and/or obesity in non-GWAS under criteria 4b [86,87,88] displayed in Appendix A. In addition, we included the 41 genes from our preliminary work that were reported to be associated with hypertension or the BP response to exercise or pharmacotherapy [27,28,29].

See SDC 1 Appendix A for the complete signature of 154 prioritized BP and exercise genes by our predetermined inclusion criteria.

## 4. Discussion

The AHA/ACC rated exercise as one of the best lifestyle therapeutic approaches to prevent and treat hypertension because aerobic exercise training lowers BP 5-8 mmHg among adults with hypertension [3]. Nearly 20 years ago, the HERITAGE study investigators established that the BP response to exercise was heritable. This finding set off investigator-initiated studies examining the genetic basis for the antihypertensive effects of exercise using a candidate gene approach, as this was the technology available at that time. Despite the best intentions and efforts of these research teams, these candidate gene studies were met with little success [17,25,26]. The failure of the candidate gene approach is best illustrated by a systematic review we recently completed on the influence of the angiotensin converting enzyme (*ACE*) insertion/deletion polymorphism rs4340 on human endurance exercise performance and cardiovascular health [89]. *ACE* rs4340 is the most widely investigated genetic variant in the exercise genomic literature. Despite the extensive volume of literature on *ACE* rs4340, we concluded that due to disparate findings no definitive conclusions could be made regarding the role of *ACE* rs4340 on endurance exercise performance or the BP response to exercise.

More recently, using newer genomic technology that emerged since our discovery phase candidate gene association studies, in a replication cohort we deep-sequenced the exons of genes contained on a prioritized signature of 41 genes identified from our earlier work combined with those from a systematic review of the literature of genes reported to be associated with hypertension or the BP response to exercise or pharmacotherapy using the Illumina TruSeq Custom Amplicon kit [27,28,29]. Even after adjustment for multiple testing, we found that 61% of the genes in the prioritized panel associated with the BP response to exercise. Clinical features such as resting BP, age, gender, and cardiometabolic biomarkers explained 66%–92% of the variation in the BP response to exercise. Yet, the genetic variants that emerged from these analyses explained 2%–15% of the variance in the BP response to exercise, a magnitude that is larger than typically reported in exercise genomic studies [25,90].

We attributed the high proportion of significant BP-genotype associations that we found to the following methodological strategies we instituted [27,28,29]: a randomized control repeated measure design with subjects who were their own control; a focused inquiry of variants with a prioritized panel of genes obtained from a systematic review of the literature that reduced the search space within the genome; high throughput exon sequencing on functional gene regions; standardized protocols that included a closely monitored, well-controlled exercise exposure; and adjustment for multiple testing based on genetic variants exhibiting variability in the number of minor alleles and with unique genotypic values. This series of studies are proof of concept that a focused genomic inquiry on functional portions of genes based upon a systematic review of the literature is a time and cost-efficient way to investigate the genomic basis for the antihypertensive effects of exercise. See Figure 2 for a conceptual overview of this systematic review approach to investigate the genomics of the antihypertensive effects of exercise. However, future investigator-initiated studies remain to be done expanding upon this approach among a large ethnically and gender diverse sample of adults with hypertension to continue to gain information about the genomic basis for the antihypertensive effects of exercise with the long-term goal of fine-tuning exercise prescriptions to optimize the BP benefits of exercise.

We acknowledge a limitation of our approach is that there are more sophisticated system analyses and bioinformatic approaches now being used with GWAS, human disease, and pathway data sets such as ENCODE and NHGRI GWAS that integrate omic high throughput technology of the genome, transcriptome, proteome, and epigenome to elucidate the genomic basis for hypertension (See SDC 1 Table 1 criteria 4a). In fact, we used such data sets in our most recent work to derive regulatory elements for the polymorphisms that passed multiple testing threshold for significant associations with the BP to exercise [27,28,29]. Another limitation is that genes have transcription-factor and post-transcriptional regulators so that RNA expression levels could also be used to find signatures associated with the antihypertensive effects of exercise as opposed to BP and exercise genes *per se*. Genes have epistatic and pleiotropic effects further complicating the identification of genetic variants that explain a clinically meaningful proportion of the BP response to exercise. Presently no data set repositories exist for genes that have been reported to associated with the BP response to exercise that have passed preestablished thresholds for multiple testing. Considering the importance of hypertension as a public health problem and the critical role exercise has in the treatment of hypertension, we posit that the methods we have presented in this systematic review represent a time and cost-efficient approach to construct targeted gene signatures whose exons can be deep-sequenced to gain insight into the genomic basis for the antihypertensive effects of exercise. Furthermore, these methods could easily be adapted to design prioritized signatures of the transcriptome, proteome, and epigenome as they regulate the BP response to exercise.

## Figures and Tables

**Figure 1 genes-10-00295-f001:**
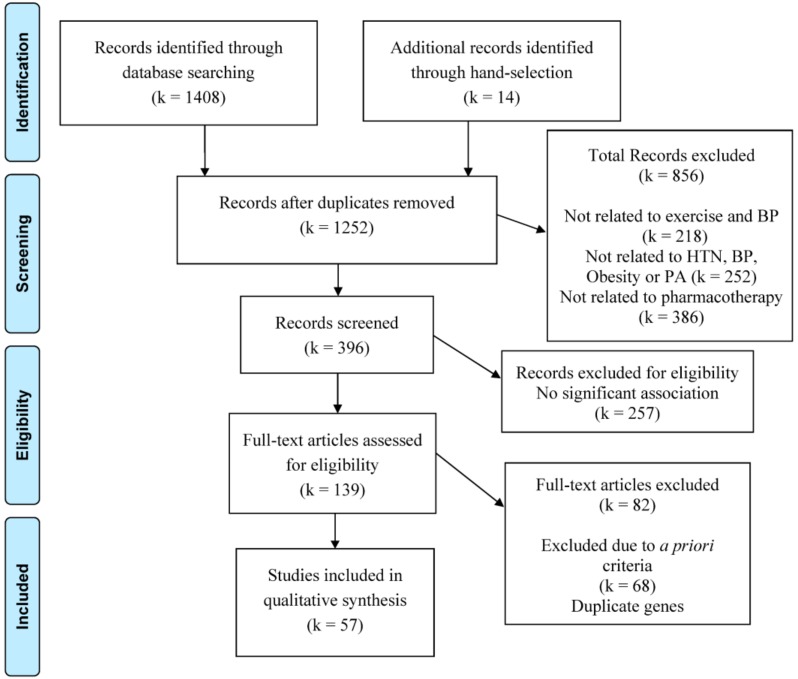
The systematic search trial selection process. BP: blood pressure; HTN: hypertension; PA: physical activity.

**Figure 2 genes-10-00295-f002:**
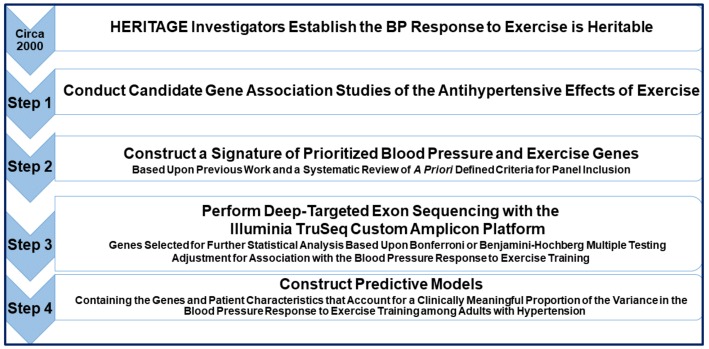
A Systematic review approach to assemble a targeted gene panel to study the genomics of the antihypertensive effects of exercise. HERITAGE: *HE*alth, *RI*sk Factors Exercise *T*r*A*ining and *Ge*netics; BP: blood pressure.

**Table 1 genes-10-00295-t001:** Criteria 1: Genes (n = 11) in major blood pressure regulatory pathways associated with the blood pressure response to exercise. Two searches were conducted on 05/11/2017 and a total of 321 records were identified.

Search 1:	(2006/01:2017/06 [edat]) AND (“mean arterial” OR “blood pressure”[mesh] OR “blood pressure” OR “blood pressures” OR “arterial pressure” OR “arterial pressures” OR hypertension OR hypotension OR normotension OR hypertensive OR hypotensive OR normotensive OR “systolic pressure” OR “diastolic pressure” OR “pulse pressure” OR “venous pressure” OR “pressure monitor” OR hypotension OR “pre hypertension” OR “bp response” OR “bp decrease” OR “bp reduction” OR “bp monitor” OR “bp monitors” OR “bp measurement”) AND (“exercise”[mesh] OR exercise OR exercises OR running[mesh] OR “bicycle” OR “bicycles” OR “bicycling” OR walking[mesh] OR treadmill* OR “weight lifting” OR “weight training” OR “weight bearing” OR “resistance training” OR “strength training” OR “endurance training” OR “speed training” OR “training duration” OR “training frequency” OR “training intensity” OR “aerobic endurance”) AND (“randomized controlled trial”[pt] OR “nonrandomized controlled” OR “nonrandomized control” OR controlled clinical trial[pt] OR “randomized controlled trial”[publication type] OR random allocation[mh] OR clinical trial[pt] OR “comparative study” OR “comparative studies” OR clinical trials[mh] OR “clinical trial”[tw] OR “latin square”[tw] OR random*[tw] OR research design[mh:noexp] OR “comparative study”[publication type] OR “evaluation studies”[publication type] OR “prospective studies”[mh] OR “cross-over studies”[mh] OR “control”[tw] OR “controlled”[tw]) AND (“gene” OR “genes” OR “genotype” OR “genotypes” OR “snp” OR polymorphism* OR “DNA” OR “minor allele” OR “minor alleles” OR “single nucleotide polymorphism” OR “single nucleotides polymorphisms” OR genetic*) NOT (“DASH”[tiab] OR “cancer” OR “neoplasms” OR “review”[pt] OR “fibromyalgia” OR “alzheimers” OR “alzheimer” OR “pregnant” OR “pregnancy” OR “obesity/drug therapy”[mesh] OR “diet therapy”[mesh] OR “diet therapy”[subheading] OR “caffeine” OR “eating change” OR “activities of daily living” OR “dehydration” OR “dehydrate” OR “dehydrated” OR “dietary salt” OR “epilepsy” OR “influenza” OR “flu” OR “pneumonia” OR “septicemia” OR “hiv” OR “Acquired Immunodeficiency Syndrome” OR “meningitis” OR “substance abuse” OR “alcoholism” OR “drug abuse” OR “Cross-Sectional Studies”[MeSH Terms] OR “Prospective Studies”[MeSH Terms] OR “epidemiology”[Subheading]).
Search 2:	(“mean arterial” OR “blood pressure”[mesh] OR “blood pressure” OR “blood pressures” OR “arterial pressure” OR “arterial pressures” OR hypertension OR hypotension OR normotension OR hypertensive OR hypotensive OR normotensive OR “systolic pressure” OR “diastolic pressure” OR “pulse pressure” OR “venous pressure” OR “pressure monitor” OR hypotension OR “pre hypertension” OR “bp response” OR “bp decrease” OR “bp reduction” OR “bp monitor” OR “bp monitors” OR “bp measurement”) AND (“ambulatory blood pressure” OR “exercise”[mesh] OR exercise[ti] OR exercises OR running[mesh] OR running[ti] OR “bicycle” OR “bicycles” OR “bicycling” OR walking[mesh] OR walking[ti] OR treadmill* OR “weight lifting” OR “weight training” OR “weight bearing” OR “resistance training” OR “strength training” OR “endurance training” OR “speed training” OR “training duration” OR “training frequency” OR “training intensity” OR “aerobic endurance” OR “aerobic training”) AND (gwa[ti] OR gwas[ti] OR genome[ti] OR “gene”[ti] OR “genes”[ti] OR “genotype”[ti] OR “genotypes”[ti] OR “genotyping”[ti] OR “snp”[ti] OR “snps”[ti] OR polymorphism*[ti] OR “DNA”[ti] OR allele[ti] OR alleles[ti] OR “minor allele” OR “minor alleles” OR “single nucleotide polymorphism” OR “single nucleotides polymorphisms” OR genetic*[ti] OR “trait locus”[ti] OR “loci”[ti] OR “Genetic Predisposition to Disease”[MeSH] OR “Genotype”[MeSH] OR “Gene Frequency”[MeSH] OR “Polymorphism, Single Nucleotide/genetics”[MESH] OR “Polymorphism, Single Nucleotide”[MAJR] OR “Genetic Loci”[Mesh] OR “Genetic Association Studies”[Mesh] OR “Genetic Variation”[Mesh]) NOT (“DASH”[tiab] OR “cancer” OR “neoplasms” OR “review”[pt] OR “fibromyalgia” OR “alzheimers” OR “alzheimer” OR “pregnant” OR “pregnancy” OR “obesity/drug therapy”[mesh] OR “diet therapy”[mesh] OR “diet therapy”[subheading] OR “caffeine” OR “eating change” OR “activities of daily living” OR “dehydration” OR “dehydrate” OR “dehydrated” OR “dietary salt” OR “epilepsy” OR “influenza” OR “flu” OR “pneumonia” OR “septicemia” OR “hiv” OR “Acquired Immunodeficiency Syndrome” OR “meningitis” OR “substance abuse” OR “alcoholism” OR “drug abuse” OR “Cross-Sectional Studies”[MeSH Terms] OR “Prospective Studies”[MeSH Terms] OR “epidemiology”[Subheading]).

**Table 2 genes-10-00295-t002:** Criteria 2: Genes (n = 18) in major blood pressure regulatory pathways associated with hypertension in genome wide association studies. The search was conducted on 05/11/2017 and a total of 188 records were identified.

(“mean arterial” OR “blood pressure”[mesh] OR “blood pressure” OR “blood pressures” OR “arterial pressure” OR “arterial pressures” OR hypertension OR hypotension OR normotension OR hypertensive OR hypotensive OR normotensive OR “systolic pressure” OR “diastolic pressure” OR “pulse pressure” OR “venous pressure” OR “pressure monitor” OR hypotension OR “pre hypertension” OR “bp response” OR “bp decrease” OR “bp reduction” OR “bp monitor” OR “bp monitors” OR “bp measurement” OR “Blood Pressure/genetics”[MeSH]) AND (“resting blood pressure” OR “resting BP” OR “exercise”[mesh] OR exercise[ti] OR exercises OR running[mesh] OR running[ti] OR “bicycle” OR “bicycles” OR “bicycling” OR walking OR walking[mesh] OR walking[ti] OR treadmill* OR “weight lifting” OR “weight training” OR “weight bearing” OR “resistance training” OR “strength training” OR “endurance training” OR “speed training” OR “training duration” OR “training frequency” OR “training intensity” OR “aerobic endurance” OR “aerobic training” OR “physical activity” OR “motor activity”[mesh] OR Overweight OR “Overweight”[Mesh] OR BMI OR “body mass” OR “Body Mass Index”[MeSH] OR “Waist Circumference”[MeSH] OR obesity OR “Obesity”[Mesh] OR obese) AND (gwa OR gwas OR “Genetic Association Studies”[Mesh] OR genomewide OR “genome wide” OR “genomewide association” OR “genome wide association” OR “genome-wide interaction”) NOT (“review”[pt] OR “Cross-Sectional Studies”[MeSH Terms] OR Comment[pt] OR Editorial[pt] OR Letter[pt] OR “Case Reports”[pt] OR “case control”[ti] OR “case report”[ti] OR “case study”[ti] OR “case series”[ti] OR “Case-Control Studies”[Mesh] OR “Follow-Up Studies”[Mesh] OR “observational study”[ti] OR “prospective cohort”[ti] OR “cohort studies” [Mesh:NoExp] OR “cohort study”[ti] OR “Longitudinal Studies” [Mesh:NoExp] OR “Follow-Up Studies”[mesh] OR “Retrospective Studies”[mesh] OR “non-randomized”[ti] OR “follow up study”[ti] OR “Cross-Sectional Studies”[MeSH Terms] OR “Prospective Studies”[MeSH Terms] OR “epidemiology”[Subheading] OR “pulmonary hypertension” OR “pulmonary arterial hypertension” OR “heart transplant” OR “heart failure” OR “cystic fibrosis” OR “cancer” OR “neoplasms” OR “fibromyalgia” OR “alzheimers” OR “alzheimer” OR “pregnant” OR “pregnancy” OR “obesity/drug therapy”[mesh] OR “diet therapy”[mesh] OR “diet therapy”[subheading] OR “DASH”[tiab] OR meal[ti] OR “nutritional intervention” OR “dietary intervention” OR “nutritional counseling” OR “dietary counseling” OR “caffeine” OR “eating change” OR “activities of daily living” OR “dehydration” OR “dehydrate” OR “dehydrated” OR “dietary salt” OR sodium OR “epilepsy” OR “influenza” OR “flu” OR “pneumonia” OR “septicemia” OR arthritis OR “hiv” OR “Acquired Immunodeficiency Syndrome” OR “meningitis” OR “substance abuse” OR “alcoholism” OR “drug abuse” OR “spinal cord”[ti] OR “Sleep”[Majr] OR “Sleep Apnea Syndromes”[Majr] OR sleep[ti] OR contraceptive*[ti] OR (animals[mesh] NOT humans[mesh]) OR rat[ti] OR rats[ti] OR mouse[ti] OR mice[ti] OR pig[ti] OR pigs[ti] OR dog[ti] OR dogs[ti] OR canine[ti] OR cow[ti] OR cows[ti] OR bovine[ti]).

**Table 3 genes-10-00295-t003:** Criteria 3. Genes (n = 28) in major blood pressure regulatory pathways associated with the blood pressure response to pharmacotherapy. The search was conducted on 05/11/2017 and a total of 711 records were identified.

(“mean arterial” OR “blood pressure”[mesh] OR “blood pressure” OR “blood pressures” OR “arterial pressure” OR “arterial pressures” OR hypertension OR hypotension OR normotension OR hypertensive OR hypotensive OR normotensive OR “systolic pressure” OR “diastolic pressure” OR “pulse pressure” OR “venous pressure” OR “pressure monitor” OR hypotension OR “pre hypertension” OR “bp response” OR “bp decrease” OR “bp reduction” OR “bp monitor” OR “bp monitors” OR “bp measurement” OR “Blood Pressure/genetics”[Mesh]) AND (“Antihypertensive Agents”[Mesh] OR “Antihypertensive Agents” [Pharmacological Action] OR “anti-hypertensive agent” OR “antihypertensive agent” OR “anti hypertensive agent” OR “anti-hypertensive agents” OR “antihypertensive agents” OR “anti hypertensive agents” OR “anti-hypertensive drug” OR “antihypertensive drug” OR “anti hypertensive drug” OR “anti-hypertensive drugs” OR “antihypertensive drugs” OR “anti hypertensive drugs” OR “anti-hypertensive medication” OR “antihypertensive medication” OR “anti hypertensive medication” OR “anti-hypertensive medications” OR “antihypertensive medications” OR “anti hypertensive medications” OR “anti-hypertensives” OR antihypertensives OR “anti hypertensives” OR diuretic OR diuretics OR “Diuretics”[Mesh] OR acebutolol OR aliskiren OR Ambrisentan OR amlodipine OR atenolol OR “azilsartan medoxomil” OR benazepril OR betaxolol OR bisoprolol OR bosentan OR “candesartan cilexetil” OR captopril OR carteolol OR carvedilol OR chlorthalidone OR clonidine OR cilazapril OR clevidipine OR deserpidine OR diazoxide OR diltiazem OR doxazosin OR enalapril OR enalaprilat OR “eprosartan mesylate” OR hydrochlorothiazide OR felodipine OR fenoldopam OR fosinopril OR guanabenz OR guanadrel OR guanethidine OR guanfacine OR hydralazine OR irbesartan OR isradipine OR labetalol OR lisinopril OR “losartan potassium” OR macitentan OR mecamylamine OR methyldopa OR metoprolol OR metyrosine OR mibefradil OR minoxidil OR moexipril OR moxonidine OR nadolol OR nebivolol OR nicardipine OR nifedipine OR nisoldipine OR nitroprusside OR “olmesartan medoxomil” OR omapatrilat OR penbutolol OR perindopril OR phentolamine OR pindolol OR prazosin OR propranolol OR quinapril OR ramipril OR rescinnamine OR reserpine OR sildenafil OR “sodium nitroprusside” OR tadalafil OR telmisartan OR terazosin OR timolol OR trandolapril OR treprostinil OR trimethaphan OR valsartan OR verapamil OR diuretic OR diuretics OR thiazide OR “adrenergic beta-antagonist” OR “adrenergic beta-antagonists” OR “adrenergic alpha-antagonist” OR “adrenergic alpha-antagonists” OR “Angiotensin-Converting Enzyme Inhibitors”[Mesh] OR “ace inhibitor” OR “ace inhibitors” OR “angiotensin-converting enzyme inhibitor” OR “angiotensin-converting enzyme inhibitors” OR “angiotensin II Receptor Blockers” OR “angiotensin II Receptor Blockers” OR “Angiotensin II Type 2 Receptor Blockers”[Mesh] OR “calcium channel blocker” OR “calcium channel blockers” OR “ganglionic blocker” OR “ganglionic blockers” OR “vasodilator agent” OR “vasodilator agents” OR “Vasodilator Agents”[Mesh] OR nitrates OR “Nitrates”[Mesh] OR nitrites OR “Nitrites”[Mesh]) AND (gwa[ti] OR gwas[ti] OR genome[ti] OR “gene”[ti] OR “genes”[ti] OR “genotype”[ti] OR “genotypes”[ti] OR “genotyping”[ti] OR “snp”[ti] OR “snps”[ti] OR polymorphism*[ti] OR “DNA”[ti] OR allele[ti] OR alleles[ti] OR “minor allele” OR “minor alleles” OR “single nucleotide polymorphism” OR “single nucleotides polymorphisms” OR genetic*[ti] OR “trait locus”[ti] OR “loci”[ti] OR “Genetic Predisposition to Disease”[MeSH] OR “Genotype”[MeSH] OR “Gene Frequency”[MeSH] OR “Polymorphism, Single Nucleotide/genetics”[MESH] OR “Polymorphism, Single Nucleotide”[MAJR] OR “Genetic Loci”[Mesh] OR “Genetic Association Studies”[Mesh] OR “Genetic Variation”[Mesh] OR “Blood Pressure/genetics”[Mesh]) NOT (“review”[pt] OR “Cross-Sectional Studies”[MeSH Terms] OR Comment[pt] OR Editorial[pt] OR Letter[pt] OR “Case Reports”[pt] OR “case control”[ti] OR “case report”[ti] OR “case study”[ti] OR “case series”[ti] OR “Case-Control Studies”[Mesh] OR “Follow-Up Studies”[Mesh] OR “observational study”[ti] OR “prospective cohort”[ti] OR “cohort studies” [Mesh:NoExp] OR “cohort study”[ti] OR “Longitudinal Studies” [Mesh:NoExp] OR “Follow-Up Studies”[mesh] OR “Retrospective Studies”[mesh] OR “non-randomized”[ti] OR “follow up study”[ti] OR “Cross-Sectional Studies”[MeSH Terms] OR “Prospective Studies”[MeSH Terms] OR “epidemiology”[Subheading] OR “pulmonary hypertension” OR “pulmonary arterial hypertension” OR “heart transplant” OR “heart failure” OR “cystic fibrosis” OR “cancer” OR “neoplasms” OR “fibromyalgia” OR “alzheimers” OR “alzheimer” OR “pregnant” OR “pregnancy” OR “obesity/drug therapy”[mesh] OR “diet therapy”[mesh] OR “diet therapy”[subheading] OR “DASH”[tiab] OR meal[ti] OR “nutritional intervention” OR “dietary intervention” OR “nutritional counseling” OR “dietary counseling” OR “caffeine” OR “eating change” OR “activities of daily living” OR “dehydration” OR “dehydrate” OR “dehydrated” OR “dietary salt” OR sodium OR “epilepsy” OR “influenza” OR “flu” OR “pneumonia” OR “septicemia” OR arthritis OR “hiv” OR “Acquired Immunodeficiency Syndrome” OR “meningitis” OR “substance abuse” OR “alcoholism” OR “drug abuse” OR “spinal cord”[ti] OR “Sleep”[Majr] OR “Sleep Apnea Syndromes”[Majr] OR sleep[ti] OR contraceptive*[ti] OR (animals[mesh] NOT humans[mesh]) OR rat[ti] OR rats[ti] OR mouse[ti] OR mice[ti] OR pig[ti] OR pigs[ti] OR dog[ti] OR dogs[ti] OR canine[ti] OR cow[ti] OR cows[ti] OR bovine[ti]).

**Table 4 genes-10-00295-t004:** Criteria 4a: Genes (n = 27) in major blood pressure regulatory pathways associated with physical activity and/or obesity in genome wide association studies. The search was conducted on 05/11/2017 and a total of 188 records were identified.

(“mean arterial” OR “blood pressure”[mesh] OR “blood pressure” OR “blood pressures” OR “arterial pressure” OR “arterial pressures” OR hypertension OR hypotension OR normotension OR hypertensive OR hypotensive OR normotensive OR “systolic pressure” OR “diastolic pressure” OR “pulse pressure” OR “venous pressure” OR “pressure monitor” OR hypotension OR “pre hypertension” OR “bp response” OR “bp decrease” OR “bp reduction” OR “bp monitor” OR “bp monitors” OR “bp measurement” OR “Blood Pressure/genetics”[MeSH]) AND (“resting blood pressure” OR “resting BP” OR “exercise”[mesh] OR exercise[ti] OR exercises OR running[mesh] OR running[ti] OR “bicycle” OR “bicycles” OR “bicycling” OR walking OR walking[mesh] OR walking[ti] OR treadmill* OR “weight lifting” OR “weight training” OR “weight bearing” OR “resistance training” OR “strength training” OR “endurance training” OR “speed training” OR “training duration” OR “training frequency” OR “training intensity” OR “aerobic endurance” OR “aerobic training” OR “physical activity” OR “motor activity”[mesh] OR Overweight OR “Overweight”[Mesh] OR BMI OR “body mass” OR “Body Mass Index”[MeSH] OR “Waist Circumference”[MeSH] OR obesity OR “Obesity”[Mesh] OR obese) AND (gwa OR gwas OR “Genetic Association Studies”[Mesh] OR genomewide OR “genome wide” OR “genomewide association” OR “genome wide association” OR “genome-wide interaction”) NOT (“review”[pt] OR “Cross-Sectional Studies”[MeSH Terms] OR Comment[pt] OR Editorial[pt] OR Letter[pt] OR “Case Reports”[pt] OR “case control”[ti] OR “case report”[ti] OR “case study”[ti] OR “case series”[ti] OR “Case-Control Studies”[Mesh] OR “Follow-Up Studies”[Mesh] OR “observational study”[ti] OR “prospective cohort”[ti] OR “cohort studies” [Mesh:NoExp] OR “cohort study”[ti] OR “Longitudinal Studies” [Mesh:NoExp] OR “Follow-Up Studies”[mesh] OR “Retrospective Studies”[mesh] OR “non-randomized”[ti] OR “follow up study”[ti] OR “Cross-Sectional Studies”[MeSH Terms] OR “Prospective Studies”[MeSH Terms] OR “epidemiology”[Subheading] OR “pulmonary hypertension” OR “pulmonary arterial hypertension” OR “heart transplant” OR “heart failure” OR “cystic fibrosis” OR “cancer” OR “neoplasms” OR “fibromyalgia” OR “alzheimers” OR “alzheimer” OR “pregnant” OR “pregnancy” OR “obesity/drug therapy”[mesh] OR “diet therapy”[mesh] OR “diet therapy”[subheading] OR “DASH”[tiab] OR meal[ti] OR “nutritional intervention” OR “dietary intervention” OR “nutritional counseling” OR “dietary counseling” OR “caffeine” OR “eating change” OR “activities of daily living” OR “dehydration” OR “dehydrate” OR “dehydrated” OR “dietary salt” OR sodium OR “epilepsy” OR “influenza” OR “flu” OR “pneumonia” OR “septicemia” OR arthritis OR “hiv” OR “Acquired Immunodeficiency Syndrome” OR “meningitis” OR “substance abuse” OR “alcoholism” OR “drug abuse” OR “spinal cord”[ti] OR “Sleep”[Majr] OR “Sleep Apnea Syndromes”[Majr] OR sleep[ti] OR contraceptive*[ti] OR (animals[mesh] NOT humans[mesh]) OR rat[ti] OR rats[ti] OR mouse[ti] OR mice[ti] OR pig[ti] OR pigs[ti] OR dog[ti] OR dogs[ti] OR canine[ti] OR cow[ti] OR cows[ti] OR bovine[ti]).

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
