# Peer review of "A Systematically Assembled Signature of Genes to be Deep-Sequenced for Their Associations with the Blood Pressure Response to Exercise"

_genes, 2019, doi:10.3390/genes10040295_

Round 1

Reviewer 1 Report

Pescatello et. al performed a systematic review with signature of genes reported by antihypertensive effects of exercise. The inclusion criteria encompassed human genes in major BP regulatory pathways associated in four selection strategies. This systematic review is very interesting and may contribute to develop the genetic predispositions and other clinical characteristics to optimize the BP benefits of exercise. However, minor points should be addressed in order to improve the systematic review.

1. In the introduction, please, describe how hypertension is diagnosed in patients (different levels).

2: The authors should give more details about the method to obtain signature of genes and deep-sequenced the exons sequencing with the Illumina TruSeq Custom Amplicon Plataform.

3. In the Figure 1 (The Systematic Search Trial Selection Process) to include the description of the symbols as HTN and explain why full-text articles were excluded (k = 82).

4. The authors should give more details about the adjustment for multiple testing as found in different points of the text as lines 62, 94.

5. In the results, would be more comprehensive if it were included which are the main pathways affected from signature of genes that were selected for each table associated with hypertension or the BP response to exercise or pharmacotherapy.

6. To include the limitations for systematic review.

Author Response

Please note I have uploaded on this document our cover letter and responses to both reviewers.

Reviewer 2 Report

GENERAL COMMENTS

The authors wrote a systematic review aiming to update and expand their original systematic review to assemble a prioritized signature of BP and exercise genes whose exons can then be deep-sequenced among a larger, more ethnically and gender diverse sample of adults with hypertension to better inform the genomic basis for the antihypertensive effects of exercise. The topic is of strong interest to both our readership and current scientific topics, but the review needs to be improved. The paper is well written, I'll list some things for you to consider:

·                    Why only used one databases, are there other important (eg.: Scopus, web of science)? For they could have found more articles that would help in the conclusions:

SPECIFIC COMMENTS

ABSTRACT

No comments.

INTRODUCTION

No comments.

SYSTEMATIC REVIEW METHODS

Why only used one databases, are there other important (eg.: Scopus, web of science)? For they could have found more articles that would help in the conclusions:

RESULTS

I suggest that the results be more detailed.

DISCUSSION

The discussion could be better presented.

CONCLUSION

I suggest you create a conclusion.

REFERENCES

Verify standardization of the journal.

I suggest improving Figure 1, as it is heavily deformed. Still in figure 1, I suggest demarcating the "n" to represent the studies and not "k".

Author Response

Please note I uploaded the cover letter and response to both reviewers under Reviewer 1 and attach her the edits in read that were made in response to the reviewers' comments.
